# Quantitative Strengthening Evaluation of Powder Metallurgy Titanium Alloys with Substitutional Zr and Interstitial O Solutes via Homogenization Heat Treatment

**DOI:** 10.3390/ma14216561

**Published:** 2021-11-01

**Authors:** Katsuyoshi Kondoh, Shota Kariya, Anak Khantachawana, Abdulaziz Alhazaa, Junko Umeda

**Affiliations:** 1Department of Composite Materials Processing, Joining and Welding Research Institute, Osaka University, 11-1 Mihogaoka, Osaka 567-0047, Ibaragi, Japan; kariya@jwri.osaka-u.ac.jp (S.K.); umedaj@jwri.osaka-u.ac.jp (J.U.); 2Department of Mechanical Engineering, King Mongkut’s University of Technology Thonburi (KMUTT), 126 Pracha Uthit Rd., Bang Mod, Thung Khru, Bangkok 10140, Thailand; anak.kha@kmutt.ac.th; 3King Abdullah Institute for Nanotechnology, King Saud University, 2454, Riyadh 11451, Saudi Arabia; aalhazaa@ksu.edu.sa; 4Tribology, Surface, and Interface Sciences (TSIS), Physics and Astronomy Department, College of Science, King Saud University, 2454, Riyadh 11451, Saudi Arabia

**Keywords:** powder metallurgy Ti-Zr-O alloy, decomposition, homogenization, water-quenching, grain refinement, solid-solution

## Abstract

The decomposition behavior of ZrO_2_ particles and uniform distribution of Zr and O solutes were investigated by employing in situ scanning electron microscope-electron backscatter diffraction (SEM-EBSD) analysis and thermogravimetric-differential thermal analysis (TG-DTA) to optimize the process conditions in preparing Ti-Zr-O alloys from the pre-mixed pure Ti powder and ZrO_2_ particles. The extruded Ti-Zr-O alloys via homogenization and water-quenching treatment were found to have a uniform distribution of Zr and O solutes in the matrix and also showed a remarkable improvement in the mechanical properties, for example, the yield stress of Ti-3 wt.% ZrO_2_ sample (1144.5 MPa) is about 2.5 times more than the amount of yield stress of pure Ti (471.4 MPa). Furthermore, the oxygen solid-solution was dominant in the yield stress increment, and the experimental data agreed well with the calculation results estimated using the Hall-Petch equation and Labusch model.

## 1. Introduction

The increased utilization of titanium in several applications such as aerospace, medicine, energy, and automotive technology is due to its high specific strength, excellent corrosion resistance, and biocompatibility [1,2,3]. Moreover, Ti alloys exhibit improved mechanical properties such as high strength-to-weight ratio, fatigue resistance at elevated temperatures, and excellent toughness by containing rare metals like alloying elements, including vanadium (V), niobium (Nb), tantalum (Ta), aluminum (Al), molybdenum (Mo), zirconium (Zr), chromium (Cr), nickel (Ni), and copper (Cu) [4,5], which help in phase stabilization and strengthening the material without impairing ductility. In general, several sustainable alloying designs and strategies in processing novel metal materials have been discussed to reduce the amount of these rare metals. For example, interstitial elements, namely oxygen (O), nitrogen (Ni), silicon (Si), hydrogen (H), and iron (Fe), have been employed as alloying elements to improve the mechanical strength of Ti materials by solid-solution hardening, grain refinement, and strong texture formation via powder metallurgy [6,7,8,9,10,11,12,13,14], which is the basis for the American Society for Testing and Materials (ASTM) mechanical strength classifications for pure Ti. Recently, both O and N elements, which were previously reported to reduce the ductility of Ti materials [15], have been used as interstitial solutes in α-Ti phases, resulting in the effective improvement of the high strength and ductility of Ti materials fabricated by a selective laser melting process [16,17]. Regarding the biomaterial applications of Ti materials, Zr is a useful alloying element because it belongs to group IV in the periodic table that is the same as Ti, owing to their very similar chemical properties and biocompatibilities [18,19]. Moreover, Zr is non-toxic and non-allergic. The Ti-Zr binary diagram suggests that the Zr atom is a complete solid-solution element in the α-Ti and β-Ti phases, and also exists as a substitutional solute in both phases to improve the mechanical strength [20,21]. It means biocompatible Ti-Zr-O ternary alloys with the synergy solid-solution strengthening effects of Zr and O atoms are suitable for medical device applications including dental implants. Several studies have extensively discussed the effects of Zr solid-solution behavior on their microstructural and mechanical properties. Zr solutes lead to α-Ti grain refinement by their Zener drag effects, resulting in enhanced mechanical strength (yield stress (YS) > 600 MPa, ultimate tensile strength (UTS) > 700 MPa), and a sufficient tensile elongation of 20% for cold-worked Ti-20 wt.% Zr alloys after hot-working and annealing treatment [22]. It has also been reported that the Zr solute distribution at the grain boundaries (GBs) obstructs grain growth by Zener drag and GB diffusion during annealing at 750 °C using transmission electron microscopy–energy dispersive X-ray analysis [23]. Recently, a quantitative strengthening mechanism evaluation of powder metallurgy Ti-Zr alloys was performed by considering the effects of α-Ti grain refinement and Zr solid-solution behavior on the yield stress increment by using the Hall–Petch equation and Labusch model, respectively [24]. It was clarified that Zr solutes have synergistic effects in improving the mechanical properties of Ti alloys by both grain refinement and solid-solution phenomena. Furthermore, the Zr solution behavior in the powder metallurgy Ti-4 wt.% Fe alloy, which is one of the α + β dual phase alloys, was investigated [25]. Zr solutes caused both α-Ti grain refinement and increased β-phase area fraction, resulting in a significantly improved YS of 0.2% for Ti-4 wt.% Fe-13 wt.% Zr alloy (1010 MPa) compared with the YS of the base Ti-4 wt.% Fe alloys (0.2% YS; 647 MPa). Therefore, several previous studies on the application of Ti-Zr alloys to dental implants have been performed to reduce the implant diameter [26].

The aim of this study is to quantitatively evaluate the strengthening mechanism of Ti-Zr-O alloys, which are fabricated from an elemental mixture of pure Ti powder and ZrO_2_ particles via powder metallurgy. First, an in-situ scanning electron microscope (SEM) observation of the decomposition behavior of ZrO_2_ particles during sintering and the subsequent diffusion behavior of Zr elements were investigated, and heat treatment experiments were employed to optimize the sintering and homogenization conditions. The microstructural investigation was conducted by SEM-electron backscatter diffraction (SEM-EBSD) analysis to measure the mean grain size and Schmid factor of the Ti-Zr-O alloys, which are necessary to calculate the strengthening increment. According to the previous studies mentioned above, it is obvious that Zr solutes play two roles in α-Ti grain refinement and solid-solution strengthening behavior. O solutes also have a solution-strengthening effect in α-Ti phases by increasing the lattice constant in the c-axis direction. In addition, the solid-solution behavior of these solutes had no effect on each other in the matrix. Therefore, it is possible to independently estimate each solution strengthening increment using the Labusch model [27] used in a previous study [28].

## 2. Materials and Methods

### 2.1. Pre-Mixing of Raw Powder

In this study, pure elemental powder of Ti (Toho Technical Service Co., Chigasaki, Kanagawa, Japan) was used as the base raw material with a mean particle size of 27.8 µm and Zr was selected as the alloying element in which ZrO_2_ fine particles (1, 2, and 3 wt.%, Kojundo Chemical Lab. Co., Sakado, Saitama, Japan) was used as the alloying additive with a mean particle size of 1.48 µm, as shown in Figure 1. The elemental mixture of pure Ti powder and ZrO_2_ particles was prepared through rocking-milling (RM-05, SEIWA Giken Co., Hiroshima, Japan) for 3.6 ks at 60 Hz with varying content of ZrO_2_ particles (1, 2, 3, and 5 wt.%). Subsequently, 200 g of yttria (Y_2_O_3_)-stabilized ZrO_2_ media balls with a diameter of 10 mm were added to 200 g of the mixture powder in a milling pot filled with argon (Ar) gas, which prevents the oxidation of the powder, to accelerate the uniform mixing process. As shown in Figure 1c1,c2, the 5 wt.% ZrO_2_ particles are uniformly dispersed on the Ti powder surface.

### 2.2. Sintering, Homogenization, and Tempering Process Conditions

The above elemental mixture was consolidated by spark plasma sintering (SPS, SPS-1030S, SPS Syntex Co., Kawasaki, Kanagawa, Japan), which is used for fabricating dense and sintered Ti samples, at a temperature of 1100 °C for 10.8 ks under a pressure of 30 MPa in a vacuum (<6 Pa). In addition, to complete the homogenization of Zr solutes in the matrix, additional heat treatment was performed at 1500 °C (in the β phase region) in an Ar atmosphere for 1.8 ks. Subsequently, the heated samples were quenched in water to prevent the agglomeration of Zr atoms during cooling. Furthermore, a tempering process was employed at 800 °C (α phase region) for 3.6 ks in a vacuum (<100 Pa) to control the Ti grain microstructures from changing from α′ to α phases.

### 2.3. Hot Extrusion

A hot extrusion process (SHP-200-450, Shibayama Kikai Co., Tsubame, Niigata, Japan) was used to fabricate fully dense Ti-ZrO_2_ materials. In this study, the samples were subjected to a pre-heating process at 850 °C (α phase region) for 300 s in an Ar gas atmosphere to ensure that the microstructures with uniformly dissolved Zr solutes formed via the above homogenization and tempering treatment maintain its properties and structure. After the preheating process, the sintered sample was immediately extruded to fabricate a rod with a diameter of 37 mm at the speed of 6 mm/s, and the calculated extrusion ratio was 13.7.

### 2.4. Materials Characterization

The surface of each sample was polished using a 4000-grit waterproof abrasive paper (Struers Japan, Shinagawa, Tokyo, Japan) to remove all scratches and was mirror-finished through buffing using Al_2_O_3_ ultra-fine particles with a diameter of 0.05 µm. Then, a chemical etching treatment was performed using a solution consisting of 1% HF, 5% HNO_3_, and 100% H_2_O (by volume fractions, Kojundo Chemical Lab. Co., Sakado, Saitama, Japan) for microstructure observation. A field-emission scanning electron microscope (FE-SEM) (JEOL, JSM-6500F, Tokyo, Japan) equipped with an energy dispersive X-ray spectroscopy (EDS) (JEOL, JED2300, Tokyo, Japan) system analysis was used to investigate the morphology, surface information, and chemical composition of microstructures and the Zr distribution in the matrix. Moreover, X-ray diffraction (XRD, Shimadzu, XRD-6100, Kyoto, Japan) analysis in the continuous scanning mode (scanning speed: 2.00 °/min) using Cu Kα radiation (λ = 0.154 nm, energy = 8.041 keV) with a tube voltage of 40 kV, tube current of 30 mA, and scanning step of 0.0200° was used to quantitatively evaluate the lattice constant changes of hexagonal close-packed (hcp) α-Ti crystals due to the solid solution of O and Zr atoms. The inverse pole figures (IPFs) of the Ti samples were obtained using the electron backscatter diffraction (EBSD) system installed in the FE-SEM for microstructural and texture analysis using a microstructure evaluation software (TSL OIM Data Collection 5.1 and OIM Analysis 5.31, Kanagawa, Japan). Electrolytic polishing (CH_3_COOH:HClO_4_ = 95:5) was applied to the specimen surface before EBSD analysis after mirror finish treatment. The tensile properties were evaluated using a universal testing machine (Shimadzu, AUTOGRAPH AG-50 kN, Kyoto, Japan) at a strain rate of 5 × 10^−4^ s^−1^, with a charge-coupled device camera system to measure the tensile strength from elongation to failure. The tensile testing specimens are of 15 mm gauge length and 3.5 mm diameter.

## 3. Results and Discussions

### 3.1. ZrO_2_ Decomposition and Zr Homogenization Behavior

To clarify the decomposition behavior of ZrO_2_ particles dispersed in the Ti matrix during sintering, thermogravimetric-differential thermal analysis (TG-DTA) was performed using Ti-5 wt.% ZrO_2_ composite with a heating rate of 20 K/min. The composite sample was prepared using SPS at 500 °C for 1.8 ks and followed by hot extrusion at 600 °C for 180 ks. This is based on the previous study [29] that mentioned that the reaction between ZrO_2_ particles and pure Ti powder occurred during vacuum sintering at temperatures above 700 °C, and SPS at 500 °C does not cause ZrO_2_ decomposition. As shown in Figure 2a, the ΔDTA profile as a function of temperature clearly indicates two endothermic peaks at approximately 780 and 890 °C. The latter corresponds to the β-Ti phase transformation, and the former possibly agrees with the decomposition phenomenon of ZrO_2_ particles. XRD analysis was also performed using the same samples after heat treatment at the temperature range of 600–1400 °C for 3.6 ks in a vacuum and subsequent quenching in water. In the XRD profiles shown in Figure 2b, the ZrO_2_ diffraction peak was detected in the samples that have not been subjected to heat treatment and those that are treated with a temperature that is less than 700 °C, and completely disappeared in the samples treated at temperatures above 800 °C, which indicates the occurrence of ZrO_2_ decomposition reaction. These results agreed with the first endothermic peak shown in the TG-DTA analysis mentioned above. In addition, it was apparent that as the heat treatment temperature increases, the α-Ti diffraction peaks gradually shifted to a lower angle, indicating that the lattice expansion of α-Ti crystals by Zr and O solutes originated from the added ZrO_2_ particles.

In situ SEM observations during heating at 800 °C were applied to the same Ti-5 wt.% ZrO_2_ sample to directly investigate the decomposition phenomenon of ZrO_2_ particles dispersed in Ti matrix, where the holding time was from 900 s to 5.4 ks with the constant magnification of 1000. As shown in Figure 3, compared to the initial sample (a1,a2) with no heat treatment and sample (b1,b2) that was heat-treated at 800 °C for 900 s, the white area around the ZrO_2_ particle expands to the matrix side; that is, the ZrO_2_ decomposition reaction starts in sample (b1). As the holding time increased from 900 s to 5.4 ks, the white area remarkably increased, and the IPF map of the SEM-EBSD analysis result (b2) indicated the grain growth inside the same white area. However, the results of the EDS analysis show that the original core part with high Zr content, corresponding to ZrO_2_ particles, remained the same even after heating at 800 °C for 5.4 ks. Moreover, the diffusion region of the Zr element is approximately 50 µm in diameter.

The changes in the a and c lattice constants were calculated based on the XRD profiles in Figure 2b using Bragg’s equation [30]. As shown in Figure 4a, the lattice constant in the a-axis has a value that is approximately the same as the constant value (0.2952–0.2954 nm) upon heating below 900 °C, and slightly increased by heat treatment from 900 to 1400 °C. On the other hand, the c lattice constant reveals a very small increase (0.4690–0.4694 nm) upon heating to 700 °C because of the absence of ZrO_2_ decomposition and the slight diffusion of Zr elements, as shown in the EDS results from Figure 4b1,b2. However, when the heat treatment temperature increased from 800 to 1400 °C, the c lattice constant drastically increased from 0.4702 to 0.4725 nm. According to the decomposition phenomenon of ZrO_2_ particles, the fast diffusion of O atoms in the α-Ti phase occurs after ZrO_2_ decomposition, and their solid-solution phenomenon causes a large lattice expansion in the c-axis direction, for example, Δc = 5.400 × 10^−4^ nm/at.% O and Δa = 0.122 × 10^−4^ nm/at.% O [28]. In addition, because Zr atoms gradually diffuse during heat treatment over 1000 °C and homogeneously dissolve in the matrix at 1400 °C, as shown in Figure 4(b3,b4), respectively, the c lattice constant increases more significantly as well as the lattice constant because the substitutional solution of Zr atoms in α-Ti crystals contributes to a small lattice expansion in the a-axis compared to the c-axis. For example, Δa = 1.501 × 10^−4^ nm/at.% Zr and Δc = 4.022 × 10^−4^ nm/at.% Zr are reported in the previous study [24].

According to the above analysis results, the optimum heat treatment condition of sintered Ti-ZrO_2_ samples for the complete decomposition of ZrO_2_ particles and Zr homogenization was determined to be 1500 °C (in the β phase region) for 1.8 ks in an Ar gas atmosphere and the subsequent water-quenching treatment.

Furthermore, acicular martensite (α′) phases were formed in the Ti alloys quenched at 1500 °C. To obtain α-Ti grains of the sintered samples, an additional tempering treatment at 800 °C for 3.6 ks in vacuum (<100 Pa) was employed in this study. For example, Figure 5 shows the optical microstructures of sintered Ti-3 wt.%ZrO_2_ material after water-quenching (Figure 5a) and tempering (Figure 5b) treatment. It is clear that the acicular α′ phases formed by quenching in (Figure 5a) completely turned into coarse equiaxed α-Ti grains (Figure 5b) after tempering at 800 °C.

### 3.2. Zr Agglomeration Behavior during Cooling after Homogenization

Figure 6 shows the results of the SEM-EDS analysis of as-sintered Ti-5 wt.% ZrO_2_ material before homogenization heat treatment. It obviously reveals the agglomeration of Zr solutes at the grain boundaries. According to Ti-ZrO_2_ pseudo-binary phase diagram [31] shown in Figure 7, the sintering temperature of 1100 °C is over β-transus temperature of Ti-1–5 wt.% ZrO_2_ materials. As mentioned above, since Zr atoms are uniformly dissolved in β-Ti phase, it is certain the Zr agglomeration phenomenon occurs during slow cooling process in vacuum after SPS. To understand this agglomeration behavior, as schematically illustrated in Figure 8a, the water-quenching from various temperatures (1500, 1200, and 1100 °C) during a furnace slow cooling after heating at 1500 °C for 1.8 ks in Ar gas atmosphere was applied to Ti-5 wt.% ZrO_2_ material sintered at 1100 °C for 10.8 ks.

As shown in SEM-EDS analysis results of Figure 8b1–b3, 1500 °C quenched sample (b1) reveals both Zr and O elements uniformly dissolved in the Ti matrix (α′ phases). The sample quenched from 1200 °C (b2) consists of the primary α and acicular fine α′ phases in the prior β. The former α grains contain a smaller amount of Zr elements compared to the latter, but the O content is larger than that in α′ phases. In 1100 °C quenched sample (b3), the growth and coarsening of primary α grains are observed, and the difference in Zr content between primary α grains and prior β phases increases a lot compared to the sample (b2). This microstructures formation mechanism as mentioned above is considered as below. In comparing the Ti-ZrO_2_ pseudo-binary phase diagram (c1), binary phase diagrams of Ti-Zr (c2) and Ti-O (c3), the temperature range of α + β dual phases of Ti-5 wt.% ZrO_2_ material is 960–1220 °C, while that in Ti-Zr and Ti-O systems is 850–870 °C and 950–1120 °C, respectively. Therefore, Zr solutes exist more stably in β phase of Ti-5 wt.% ZrO_2_ material. In addition, O solid-solution is also more stable in α phase in α + β temperature range (in particular higher temperature). As a result, acicular α′ grains transformed from β phases contain a larger amount of Zr solutes, and the primary α phases are an enriched area of O atoms.

According to the above investigation results, the Zr agglomeration phenomenon of the Ti-ZrO_2_ material during cooling after SPS in the β phase region is schematically illustrated in Figure 9. During heating to the β phase temperature during sintering, ZrO_2_ decomposition and diffusion of Zr and O atoms occur. Both elements were uniformly dissolved in the β-Ti phases during sintering (Figure 9a). In the α + β temperature range during cooling after sintering, the diffusion of Zr and O atoms occurs in β-Ti and primary α-Ti phases, respectively. With decreasing temperature during slow cooling, α grain growth and coarsening occurred, and Zr solutes were exhaled from the primary α-Ti to the residual β phases (Figure 9b). Finally, in the α-Ti phase region, Zr agglomerates at the prior β grain boundaries (Figure 9c), which corresponds well to the network-structured Zr distribution shown in the EDS analysis results.

### 3.3. Role of Homogenization on Microstructural and Mechanical Properties of Extruded Ti-Zr-O Alloys

The homogenized, quenched, and tempered Ti-ZrO_2_ sintered materials showed a uniform solution of Zr and O atoms in the α-Ti grains. EDS analysis of Zr distribution and micro-hardness testing were carried out to investigate the effect of these secondary operations after the SPS process on the microstructures and mechanical properties of extruded Ti-Zr-O alloys. Figure 10 shows the SEM-EDS analysis results of the extruded Ti-3 wt.% ZrO_2_ materials. When no secondary operation was applied to the sintered sample (Figure 10a1, No-H.T.), a large amount of striped Zr agglomeration area was observed along the extrusion direction, which is very similar to the microstructure of the extruded Ti-ZrH_2_ alloy with no additional heat treatment [24]. Meanwhile, the extruded sample with the secondary operation (Figure 10a2, H.T.) reveals no Zr agglomeration in the matrix. EDS point analysis (nine points) on both samples was performed perpendicular to the extrusion direction, as shown in Figure 10b1,b2. The average Zr content of both samples (Zr: 1.21–1.22 at.%) is approximately equal to the stoichiometric value (Zr; 1.19 at.%). The striped Zr concentrated area, however, has a significantly high Zr content (~1.73 at.%), which is about 1.5 times the average value. Comparing the standard deviation (SD) of each Zr measurement, the SD of sample shown in Figure 10b2 with the additional heat treatment of 0.04 at.% is much smaller than that of sample shown in Figure 10b1 with 0.22 at.% SD. That is, the secondary operation is effective for the uniform distribution of Zr solutes in the matrix even after the hot extrusion process.

Comparing the EDSB analysis results shown in Figure 11a,b, Zr solutes effectively obstruct the coarsening phenomenon of α-Ti grains, as reported in previous studies [22,24]. In addition, there is no remarkable difference in the α-Ti grain size and morphology of extruded pure Ti materials between non-homogenization and homogenization in (a). However, the secondary operation significantly contributed to the formation of fine equiaxed α-Ti grains of the extruded Ti-3 wt.% ZrO_2_ material via dynamic recrystallization in Figure 11b2, having a uniform grain size distribution compared to the sample with no heat treatment (Figure 11b1). Figure 11c shows the SEM microstructure and Zr mapping by EDS analysis of the non-homogenized Ti-3 wt.% ZrO_2_ material after chemical etching treatment. This clearly indicates that recrystallized fine α-Ti grains formed along the striped Zr agglomeration area, and coarse grains exist in the region with a relatively low Zr solute content. The nano-indentation hardness test of the same sample also showed a hardness of 4.14 GPa (Figure 11d1) and 6.80 GPa (Figure 11d2) at relatively low Zr (1.13 at.%) and high Zr (2.15 at.%) content area, respectively. A larger strain is required in the plastic deformation of the harder matrix during extrusion, which results in the acceleration of the dynamic recrystallization phenomena of α-Ti grains, causing the formation of refined grains [24]. Accordingly, fine Ti grains were formed along the striped Zr agglomeration regions with a relatively high hardness, as shown in Figure 11c.

The effect of the uniform Zr solution on the matrix hardness of the extruded Ti-3 wt.% ZrO_2_ material was carried out by using nano-indentation tester. As shown in Figure 12a, the hardness measurement (number of tests; 100) of the homogenized sample (●: H.T) is 6.02 ± 0.48 GPa, and that of non-homogenized one (◇: Non H.T) is 5.55 ± 0.84 GPa. The measurement scatter of the former is significantly controlled owing to the uniform distribution of Zr solutes by the secondary homogenization heat treatment, as shown in Figure 10a2. When the ZrO_2_ content was changed from to 1–3 wt.% shown in Figure 12b, the hardness of both samples with H.T and non H.T increases linearly with their contents. However, it is obvious that the scatter of homogenized samples is also much smaller than that in non-H.T when changing the ZrO_2_ content.

### 3.4. Quantitative Evaluation of Strengthening Mechanism of Ti-Zr-O Alloys

Figure 13 shows the tensile stress–strain curves of Ti-0-3 wt.% ZrO_2_ extruded materials with homogenization at room temperature, where the tensile properties are summarized in the attached table. With increasing ZrO_2_ content, 0.2 the yield stress (YS) and ultimate tensile strength (UTS) significantly increased; for example, the 0.2%YS mean value of Ti-3 wt.% ZrO_2_ sample (1144.5 MPa) is about 2.5 times that of pure Ti (471.4 MPa). On the other hand, the failure to elongate is sufficiently large (25.6–28.3%) until 2 wt.% ZrO_2_ addition, and even Ti-3 wt.% ZrO_2_ sample still shows 7.8% elongation.

The analysis results of the chemical compositions and microstructural and mechanical properties of extruded Ti-Zr-O alloys are summarized in Table 1, where the mean grain size and Schmid factor (S_f_) were measured by EBSD analysis. The experimental yield stress (YS) increment, $\Delta \sigma_{YS\left[E\right]}$, is the difference in YS between pure Ti (standard material) and each Ti-Zr-O alloy. According to the microstructural analysis results mentioned above, it is certain that the main strengthening factors of the extruded Ti-Zr-O alloys are α-Ti grain refinement and solid solution of Zr and O atoms. First of all, the grain boundary strengthening increment by grains refinement, $\Delta \sigma_{YS\left[GR\right]}$ can be calculated using the Hall–Petch relationship:$\;\Delta \sigma_{YS\left[GR\right]}=K\left(d_{0}^{-0.5}-d^{-0.5}\right)$ [32,33], where $d_{0}$ and d are the mean grain sizes of the pure Ti and each Ti–Zr-O alloy specimen, respectively, and *K* is the Hall–Petch constant (18.6 MPa/mm^−0.5^) [15]. As shown in Table 1, the addition of 1 wt.% ZrO_2_ particles causes a drastic decrease of α-Ti grain size and results in $\Delta \sigma_{YS\left[GR\right]}$ of 44.6 MPa. However, the further addition of ZrO_2_ was not effective in refining α-Ti grains, and there was no significant change in $\Delta \sigma_{YS\left[GR\right]}$ (36.0–50.9 MPa). On the other hand, YS increases due to Zr, O, and N atom solid solutions, $\Delta \sigma_{YS\left[Zr-SS\right]},\Delta \sigma_{YS\left[O-SS\right]}$, and $\Delta \sigma_{YS\left[N-SS\right]}$ were estimated using the Labusch model [27]:
(1)$$\Delta \sigma_{\left[SS\right]}=\frac{\tau }{S_{F}}=\frac{1}{S_{F}}{\left(\frac{F_{m}^{4}c^{2}w}{4Gb^{9}}\right)}^{1/3}$$
where c is the solute concentration, $S_{F}$ is the Schmid factor, G is the shear modulus of 4.50 × 10^10^ N/m^2^, b is the Burgers vector of 2.951 × 10^−10^ m, and w is the width of the edge dislocation equal to 5b (~1.476 × 10^−9^ m) [34]. The maximum interaction force ($F_{m}$) between each solute and dislocations in α-Ti crystals is necessary to calculate the YS increment by solid solution, $\Delta \sigma_{\left[O-SS\right]}$, using the Labusch model. For example, the $F_{m}$ value of Ti-Zr, Ti-O and Ti-N is considered to be 1.38 × 10^−10^ N [24], 6.22 × 10^−10^ N, and 5.21 × 10^−10^ N [28], respectively. The calculated YS increment for each solute is also shown in Table. The $\Delta \sigma_{\left[N-SS\right]}$ of each sample showed a negative value because the N content decreased slightly compared to the standard material of pure Ti. In comparing the strengthening factors by grain refinement and solid solution quantitatively, it is obvious that the O solid solution is the most dominant mechanism of extruded Ti-Zr-O alloys fabricated by powder metallurgy in this study. In addition, the total calculated YS increment of each sample, $\Delta \sigma_{YS\left[C\right]}$, expressed by Equation (2) is summarized in Table 1.
(2)$$\Delta \sigma_{YS\left[C\right]}=\Delta \sigma_{\left[GR\right]}+\Delta \sigma_{\left[Zr-SS\right]}+\Delta \sigma_{\left[O-SS\right]}+\Delta \sigma_{\left[N-SS\right]}$$

Figure 14 indicates the relationship of the YS increment between the experiment by tensile test ($\Delta \sigma_{YS\left[C\right]}$) and the calculation using the Hall–Petch and Labusch equations *(*$\Delta \sigma_{YS\left[E\right]}$). It shows a sufficiently high determining factor (*R*^2^) of 0.9957, which indicates a strong positive correlation between the experimental results and the calculated values. That is, it is possible to quantitatively estimate the strengthening increment by α-Ti grain refinement and each solid-solution behavior of Ti-Zr-O alloys using the microstructural analysis results, Hall–Petch equation, and Labusch model.

## 4. Conclusions

A quantitative evaluation was carried out in this study in which Ti-Zr-O alloys were fabricated from an elemental mixture of pure Ti powder and ZrO_2_ particles via powder metallurgy. It was found that the combination of the homogenization heat treatment at the β-Ti phase region (1400–1500 °C) and subsequent water quenching was effective in preventing the agglomeration of Zr solutes at prior β grain boundaries in the matrix after the decomposition of ZrO_2_ particles during sintering. When employing the pre-mixed powder of Ti-2 wt.% ZrO_2_, a good balance between 1042.8 MPa UTS and 26.4% elongation was obtained at room temperature. Moreover, the quantitative comparison of the main strengthening factors of Ti-Zr-O alloys, which was estimated by using the Hall–Petch equation and Labusch model, shows that the oxygen solid-solution was dominant in the YS increment. For example, in case of 3 wt.% ZrO_2_ particles addition, the amount of oxygen solid solution strengthening accounts for 83% of the total amount of YS increment. In addition, the experimental data agreed well with the calculation results for the total YS increment.

## Figures and Tables

**Figure 1 materials-14-06561-f001:**
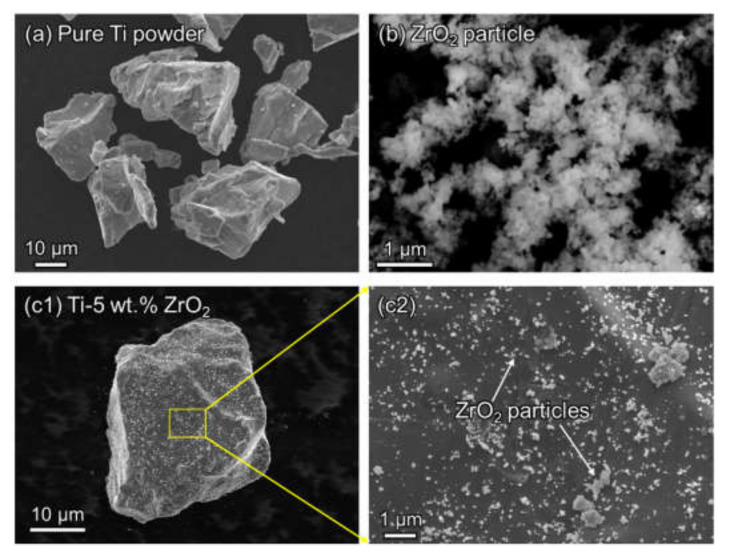
(**a**) SEM observation on pure Ti powder, (**b**) ZrO_2_ particles, (**c****1**) pre-mixed Ti-5 wt.% ZrO_2_ powder and (**c2**) ZrO_2_ particles uniformly dispersed at Ti powder surface.

**Figure 2 materials-14-06561-f002:**
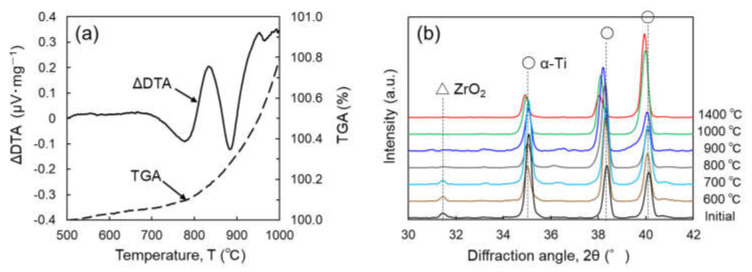
(**a**) Thermogravimetric-differential thermal analysis (TG-DTA) profiles of pre-mixed Ti-5 wt.% ZrO_2_ powder and (**b**) X-ray diffraction (XRD) profiles of Ti-5 wt.% ZrO_2_ composite sintered at 500 °C and heat treated at 600–1400 °C.

**Figure 3 materials-14-06561-f003:**
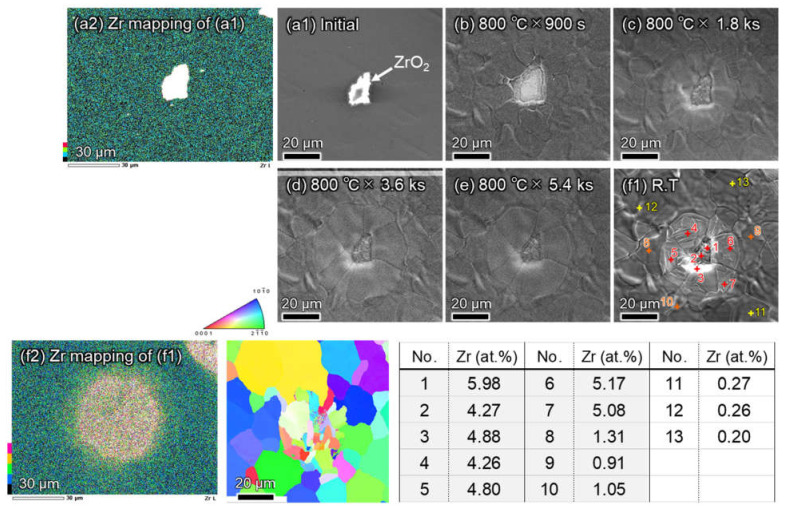
In-situ scanning electron microscope (SEM) with energy dispersive X-ray spectroscopy (EDS) and electron backscatter diffraction (EBSD) analysis results of sintered Ti-5 wt.% ZrO_2_ composite during heating at 800 °C for 900 s–5.4 ks. SEM photos of (**a1**) initial (as-sintered), heat treated at 800 ℃ for (**b**) 900 s, (**c**) 1.8 ks, (**d**) 3.6 ks, (**e**) 5.4 ks, and (**f1**) cooled to room temperature after heat treatment at 800 ℃. (**a2**) EDS result of Zr element distribution of initial sample (**a1**) and (**f2**) EDS-EBSD result of sample (f1). Attached table showing Zr measurements by ESD point analysis at No. 1–13 of sample (**f1**).

**Figure 4 materials-14-06561-f004:**
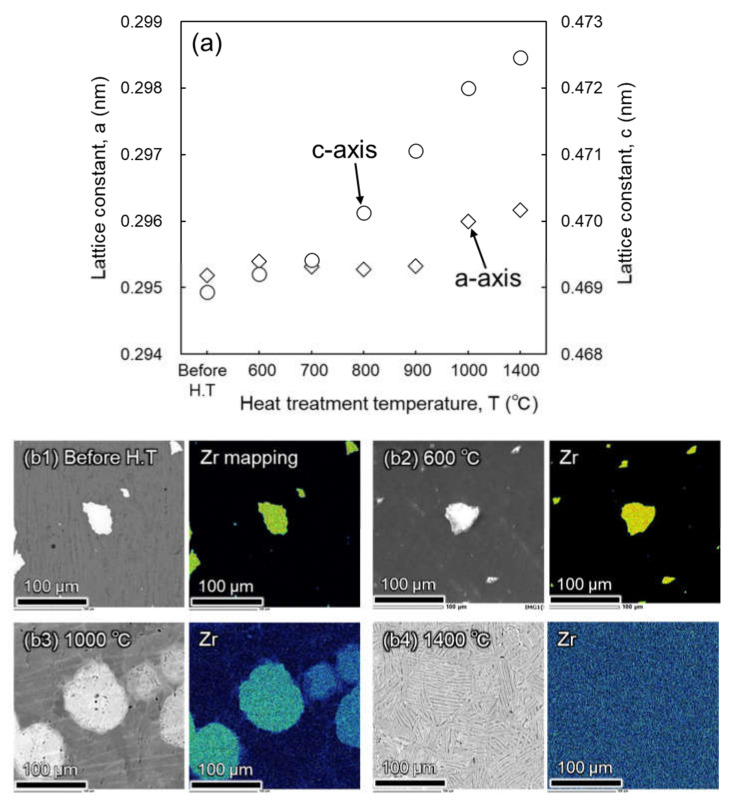
Dependence of a and c lattice constants of sintered Ti-5 wt.% ZrO_2_ composite on heat treatment temperature (**a**) and SEM-EDS analysis results of as-sintered before heat treatment (**b1**), water-quenched after heat treatment at 600 °C (**b2**), 1000 °C (**b3**), and 1400 °C (**b4**) for 3.6 ks in vacuum.

**Figure 5 materials-14-06561-f005:**
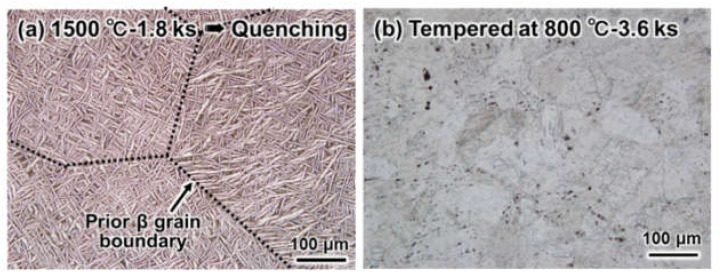
Optical microstructures of as-quenched Ti-3 wt.% ZrO_2_ material with homogenization at 1500 °C for 1.8 ks in vacuum (**a**) and subsequently tempered sample at 800 °C for 3.6 ks in vacuum (**b**).

**Figure 6 materials-14-06561-f006:**
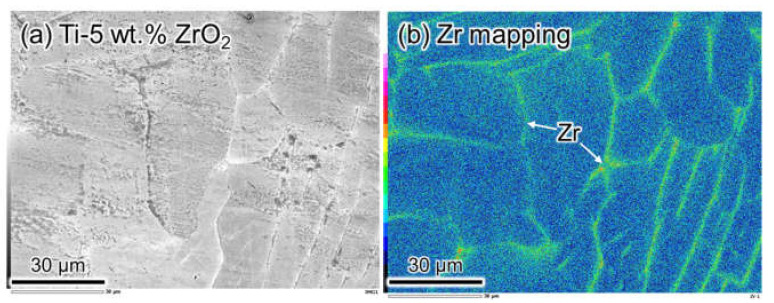
(**a**) SEM observation and (**b**) Zr mapping by EDS analysis of non-homogenized Ti-3 wt.% ZrO_2_ sintered material with Zr agglomeration at grain boundaries.

**Figure 7 materials-14-06561-f007:**
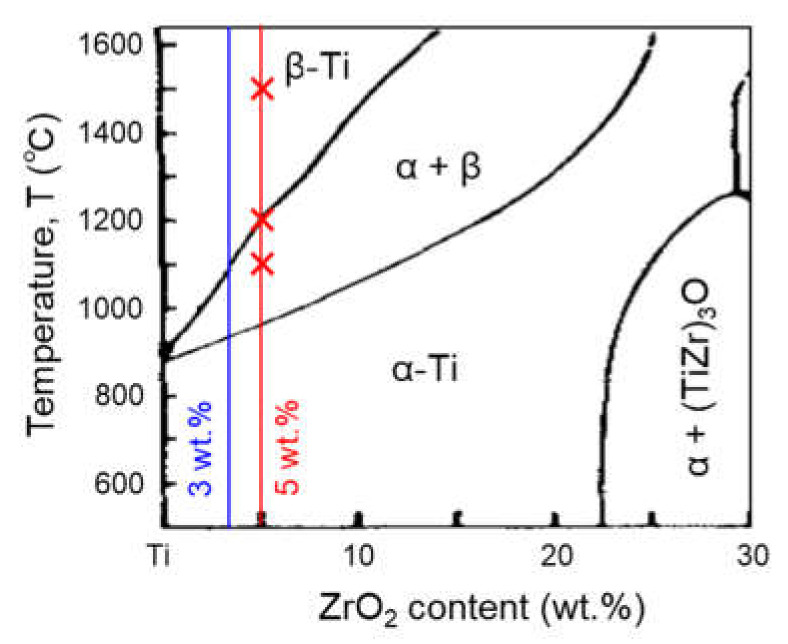
Ti-ZrO_2_ pseudo-binary phase diagram [31].

**Figure 8 materials-14-06561-f008:**
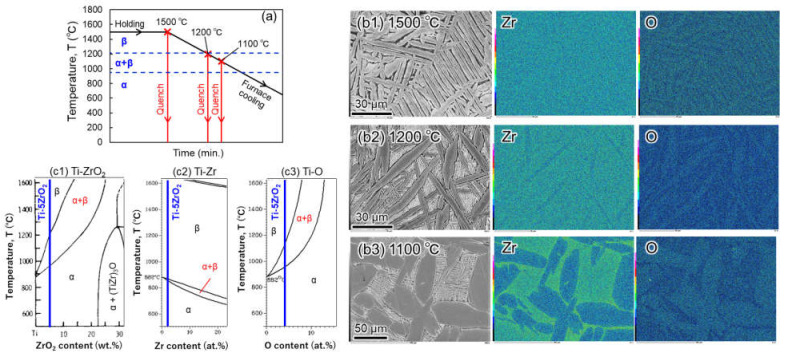
(**a**) Schematic illustration of water-quenching experiment conditions. SEM-EDS analysis results of Ti-5 wt.% ZrO_2_ material after quenching from (**b1**) 1500 ℃, (**b2**) 1200 ℃, (**b3**) 1100 °C, and binary phase diagrams of (**c1**) Ti-ZrO_2_, (**c2**) Ti-Zr and (**c3**) Ti-O.

**Figure 9 materials-14-06561-f009:**
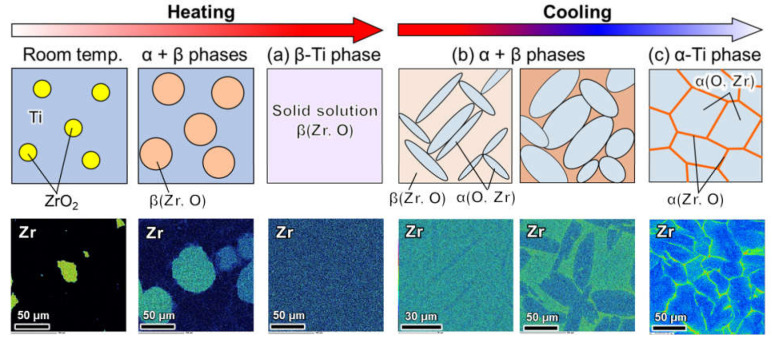
Schematic illustration of Zr agglomeration behavior during cooling after sintering at β phase region in employing pre-mixed pure Ti and ZrO_2_ powder as starting materials, and microstructure changes observed by SEM-EDS analysis. (**a**) β-Ti single phase region during heating, (**b**) α + β dual phase and (**c**) α-Ti single phase regions during cooling in sintering.

**Figure 10 materials-14-06561-f010:**
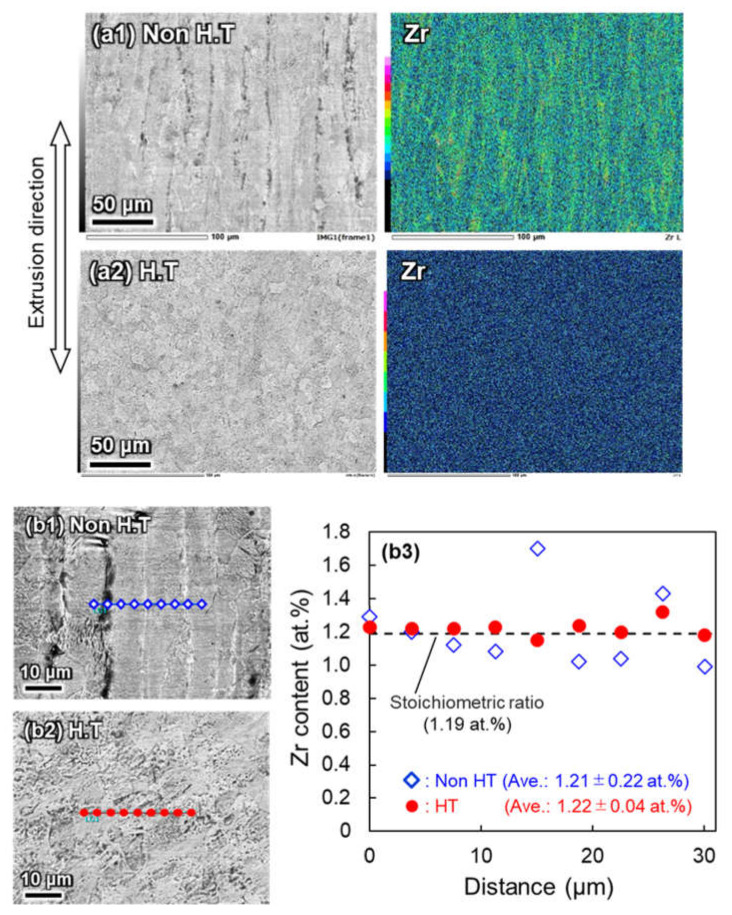
SEM-EDS analysis results of extruded Ti-3 wt.% ZrO_2_ alloy with (**a1**) no homogenization and (**a2**) homogenization, and (**b1**–**b3**) Zr content distribution of each Ti-Zr-O alloy.

**Figure 11 materials-14-06561-f011:**
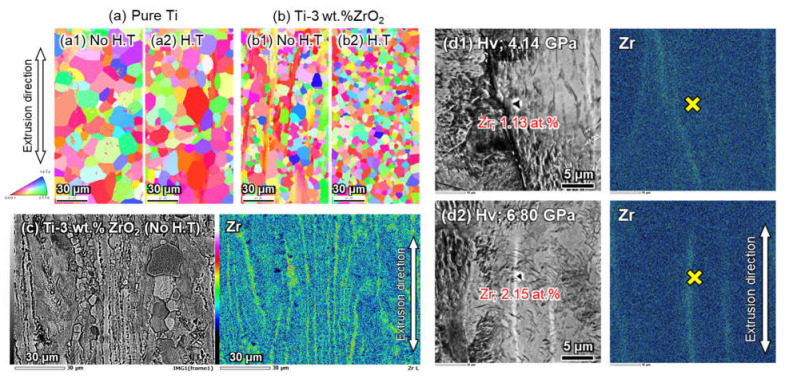
(**a**) EBSD analysis on pure Ti and (**b**) Ti-3 wt.% ZrO_2_ material (1) with or (2) without homogenization, (**c**) SEM-EDS results of extruded Ti-3 wt.% ZrO_2_ material with non-homogenization, and (**d1**,**d2**) relationship between Zr content and micro-hardness measured by nano-indentation of non-heat treated sample (**c**).

**Figure 12 materials-14-06561-f012:**
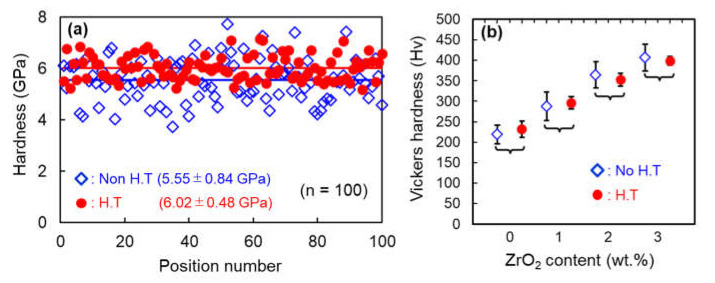
(**a**) Micro-hardness distribution of extruded Ti-3 wt.% ZrO_2_ materials with or without homogenization and (**b**) dependence of hardness measurement on ZrO_2_ content of extruded composites with or without homogenization.

**Figure 13 materials-14-06561-f013:**
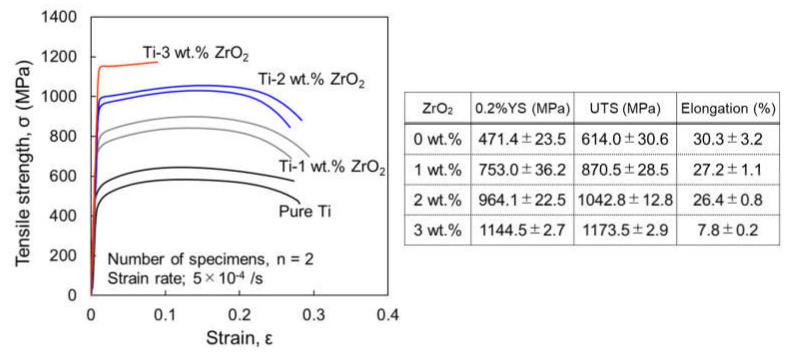
Tensile stress–strain curves of Ti-x wt.% ZrO_2_ (x = 0, 1, 2, 3) materials with homogenization heat treatment at room temperature and table showing summarized tensile properties of each sample.

**Figure 14 materials-14-06561-f014:**
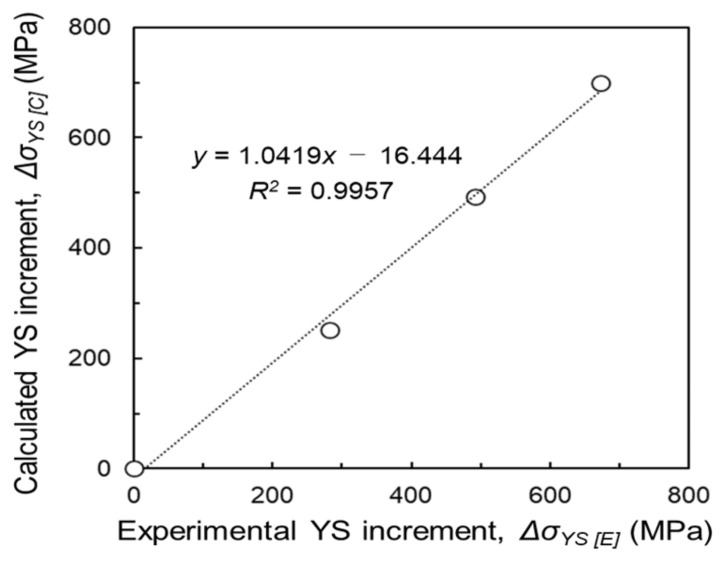
Relationship of yield stress increment between experiment by tensile test ($\Delta \sigma_{YS\left[E\right]}$) and calculation by Hall–Petch and Labusch equations ($\Delta \sigma_{YS\left[C\right]}$ ).

**Table 1 materials-14-06561-t001:** Summaries of compositions of each solution element, microstructures characteristics, experimental tensile properties of extruded Ti-Zr-O alloys with homogenization and their calculated YS increments by each strengthening factor.

ZrO_2_ (wt.%)	Chemical Compositions (at.%)				Microstructural and Mechanical Properties						Calculated Stress Increment/MPa				
	Zr	O	N	H	Grain Size (µm)	Schmid Factor, *S_F_*	YS, *σ_YS_* (MPa)	UTS, *σ*(MPa)	Elongation (%)	YS increment, Δ*σ**_YS[E]_* (MPa)	Δ*σ_[GR]_*	Δ*σ_[Zr-SS]_*	Δ*σ_[O-SS]_*	Δ*σ_[N-SS]_*	YS Increment, Δ*σ_YS[C]_*
0	0	1.08	0.07	0.79	13.51	0.43	471.4	614.0	30.3	0	0	0	0	0	0
1.0	0.39	1.82	0.06	0.88	8.14	0.44	753.0	870.5	27.2	281.6	44.6	31.6	181.7	−7.1	250.8
2.0	0.79	2.65	0.06	0.93	8.89	0.42	964.1	1042.8	26.4	492.7	36.0	52.7	408.3	−4.6	492.4
3.0	1.19	3.34	0.06	0.97	7.65	0.41	1144.5	1173.5	7.8	673.1	50.9	71.0	580.4	−3.2	699.1

## Data Availability

The data presented in this study are available on request from the corresponding author.

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
