# Peer review of "Quantitative Strengthening Evaluation of Powder Metallurgy Titanium Alloys with Substitutional Zr and Interstitial O Solutes via Homogenization Heat Treatment"

_materials, 2021, doi:10.3390/ma14216561_

Round 1

Reviewer 1 Report

The article is interesting and well presented.

There still some aspects that should be corrected before publication:

Figure 1. I don't think is compulsory. 

Figure 3 - should be presented larger - add maping and table bellow the microimages. 

Figure 4. same as Figure 3. 

On Figure 6 - on mapping, Zr what color is it? blue/yellow.. a legend should be there.

Add more current references (2020 – 2021) about powder metallurgy.

Improve the introduction with data about titanium alloys from the literature. Why you choose this alloy? Which is the novelty of your paper? Discuss about the applications of this type of alloy.

Example of references:

https://link.springer.com/article/10.1140%2Fepjp%2Fs13360-021-01590-x
https://doi.org/10.3390/met9101033
https://doi.org/10.3390/met9010076

 Add more detailed conclusions.  - Generally the quality of the writing could be improved.

Author Response

Figure 1. I don't think is compulsory. 

âž The pre-mixed Ti-ZrO2 powder was used as the starting materials in this study. It is important to uniformly disperse fine ZrO2 particles in coarse Ti powder to obtain Ti-Zr-O sintered alloys with no agglomeration of Zr and O elements. Fig. 1 (c) is necessary to show ZrO2 particles uniformly exist on Ti power surface after ball-milling process. Therefore, I hope you understand Figure 1 is compulsory in this manuscript. In addition, the scale bars were clearly added in each figure.

Figure 3 - should be presented larger - add maping and table bellow the microimages. 

âž  Fig. 3 was presented larger to clearly show the analysis results by SEM-EDS. The scale bars were clearly added in each figure. EDS mapping results and table showing point analysis results were move below SEM images as you suggested.

Figure 4. same as Figure 3. 

âž  As you suggested, SEM-EDS results were moved below the graph of the lattice constant changes.

On Figure 6 - on mapping, Zr what color is it? blue/yellow.. a legend should be there.

âž  Zr mapping result was revised to show where Zr elements exist in the matrix.

Add more current references (2020 – 2021) about powder metallurgy.

âž  As you suggested, more current papers regarding PM Ti alloys were employed in the text and references.

Improve the introduction with data about titanium alloys from the literature. Why you choose this alloy? Which is the novelty of your paper? Discuss about the applications of this type of alloy.

âž  The features and merits of Zr and O solutes in Ti alloys are explained in the text, and then the biocompatible Ti-Zr-O ternary alloy with high strength can be applied to the medical devices including dental implants. These are included in the revised manuscript. The novelty of this article, which is not included in the text, is as follows;

The decomposition behavior of ZrO2 in Ti matrix during sintering was investigated, and the martensite formation mechanism with uniform Zr distribution by Zr solutes was also clarified. The synergy solid-solution strengthening effects of Zr and O atoms on Ti alloys were quantitatively evaluated by using Hall-Petch equation and Labusch model.

Reviewer 2 Report

Excellent paper. Can be published as it is.

Minor: Please add scale bars to Fig.1

Author Response

Minor: Please add scale bars to Fig.1

âž  As suggested by a reviewer, the scale bars were clearly added in each figure.

Reviewer 3 Report

In the manuscript “Quantitative Strengthening Evaluation of Powder Metallurgy Titanium Alloys with Substitutional Zr And Interstitial O Solutes Via Homogenization Heat Treatment”, the authors investigated the decomposition behavior of ZrO2 particles and the distribution of Zr and O, as well as the effect of the Zr and O concentrations on the mechanical properties of titanium alloys. The authors used modern methods for analysis such as the in situ scanning electron microscope-electron backscatter diffraction (SEM-EBSD) analysis and thermogravimetric-differential thermal analysis (TG-DTA). All in all, this is an excellent manuscript. The authors analyzed in detail the microstructure evolution and explained all the phase transformations. The conclusions made by the authors are fully supported by the investigation results.

I have a few small comments on the manuscript:

  1. The Materials and Methods do not contain information about the dimensions of the tensile test samples.
  2. The scale marks in Figures 1 and 3 are completely invisible.

Author Response

I have a few small comments on the manuscript:

  1. The Materials and Methods do not contain information about the dimensions of the tensile test samples.

âž  The below sentence is added in the text.

The tensile testing specimens are of 15 mm gauge length and 3.5 mm diameter.

  1. The scale marks in Figures 1 and 3 are completely invisible.

âž  As you suggest, the scale bars were clearly added in both figures.